# Heart Failure and Drug Therapies: A Metabolic Review

**DOI:** 10.3390/ijms23062960

**Published:** 2022-03-09

**Authors:** Frank Yu, Bianca McLean, Mitesh Badiwala, Filio Billia

**Affiliations:** 1Peter Munk Cardiac Centre, Toronto General Hospital, Toronto, ON M5G 2C4, Canada; frank.yu@uhnresearch.ca (F.Y.); mitesh.badiwala@uhn.ca (M.B.); 2Ted Rogers Centre for Heart Research, Toronto, ON M5G 1X8, Canada; 3Degroote School of Medicine, McMaster University, Hamilton, ON L8S 4L8, Canada; mcleanbi@hhsc.ca; 4Toronto General Hospital Research Institute, Toronto, ON M5G 2C4, Canada

**Keywords:** metabolism, heart failure, pharmacology

## Abstract

Cardiovascular disease is the leading cause of mortality globally with at least 26 million people worldwide living with heart failure (HF). Metabolism has been an active area of investigation in the setting of HF since the heart demands a high rate of ATP turnover to maintain homeostasis. With the advent of -omic technologies, specifically metabolomics and lipidomics, HF pathologies have been better characterized with unbiased and holistic approaches. These techniques have identified novel pathways in our understanding of progression of HF and potential points of intervention. Furthermore, sodium-glucose transport protein 2 inhibitors, a drug that has changed the dogma of HF treatment, has one of the strongest types of evidence for a potential metabolic mechanism of action. This review will highlight cardiac metabolism in both the healthy and failing heart and then discuss the metabolic effects of heart failure drugs.

## 1. Introduction

Cardiovascular disease (CVD) has the highest healthcare burden worldwide and continues to increase annually [1,2,3,4]. Heart failure (HF) is a significant contributor of CVD mortality and is a complex syndrome that can range from minimal loss of functional capacity to severe symptoms refractory to therapy [5,6]. While our knowledge of this disease has rapidly evolved, current treatment strategies are limited to neurohormonal inhibition, targeting of the sympathetic nervous system and reduction in myocardial oxygen consumption through ventricular unloading (Table 1) [7,8]. In contrast, there is a lack of agents targeting cardiac metabolism.

HF has been associated with altered substrate utilization and oxidative stress, ultimately leading to an energy deficit [9,10,11]. Advances in ‘-omics’ approaches have better defined the metabolic alterations in the heart. However, translating these findings into therapeutic strategies has been challenging because of the complex control of energy metabolism that is closely intertwined with an extensive signaling network. In this review, we examine the main pathways of cardiac energy metabolism and the associated metabolic shift in a failing heart. Finally, we will discuss how standard and novel HF treatments can potentially target metabolic pathways.

## 2. Metabolism in the Healthy Heart

### 2.1. Substrate Utilization

The heart hydrolyzes 20 times its mass in adenosine triphosphate (ATP) every day, but it has a limited ATP reserve lasting only seconds without replenishment. Thus, it is not surprising that perturbations in the heart’s ability to generate energy or utilize substrates can have profound impact on its function [12,13,14]. The energy pool of the healthy heart consists of ~5 μmol of ATP per gram of wet weight and allows the heart to meet the cardiovascular demands of 50 mL/kg/min VO_2_ delivery during maximal exercise (VO_2_ max) [15,16]. >95% of the heart’s ATP is generated from oxidative phosphorylation in the mitochondria, while the remaining 5% is from glycolysis and to a lesser extent, the tricarboxylic acid (TCA) cycle [17]. About 60% to 70% of the heart’s ATP is utilized for contractile proteins, predominantly the myosin ATPase [18]. The remaining 30 to 40% of myocardial ATP is used to operate various ion pumps, especially the sarcoplasmic reticulum Ca^2+^-ATPase (SERCA) and the Na^+^/Ca^+^ exchanger 1 (NX1) [19,20]. SERCA and NX1 are both ion pumps that maintain intracellular calcium ion concentrations for excitation-contraction coupling.

The heart can metabolize fatty acids, glucose, ketone bodies, and amino acids as required. Under normoxic conditions, the heart depends on fatty acid (FA) oxidation to produce 60–90% of its ATP [10]. The remaining 10–40% of ATP is generated from the oxidation of pyruvate, whereby glycolysis and lactate oxidation contribute equal amounts of pyruvate. The heart’s preference of FA over glucose can be attributed to ATP production efficiency. One molecule of palmitate yields a net gain of 129 ATP and consumes 31 molecules of oxygen. One molecule of glucose yields 38 ATP and consumes 6 molecules of oxygen. Hence, FAs produce significantly more ATP than glucose under normoxic conditions. Ketone bodies and amino acids represent a less significant source of energy under physiological conditions, but the relative substrate utilization can rapidly change due to demand and altered conditions. Ketone bodies can become the major source of energy during prolonged fasting or poorly controlled diabetes. Furthermore, the heart will switch from FA to glucose under hypoxic conditions [21]. The heart’s flexible substrate utilization allows it to adapt to its environment and maintain function [22].

### 2.2. Substrate Selection

The heart’s ability to rapidly switch between substrates is determined by the catalytic activity and bioavailability of several rate-limiting enzymes [7]. These processes are regulated both at the transcriptional and translational level in response to the cellular environment. For instance, the relative contributions of FA and glucose is regulated by the glucose-fatty acid cycle, otherwise known as the Randle cycle [23]. In this process, glucose metabolism is impaired through the phosphorylation of pyruvate dehydrogenase (PDH) and 6-phosphofructo-1-kinase (PFK-1), the rate-limiting enzymes involved in glucose oxidation (Figure 1) [24].

FA oxidation inhibits PDH activity through the activation of mitochondrial pyruvate dehydrogenase kinase (PDK) resulting in increased ratios of acetyl coenzyme A (CoA)/CoA and NADH/NAD^+^. While PDH is completely inhibited by FAs, PFK exhibits a relatively smaller effect where 40–60% of its activity is limited [25]. However, FA inhibition of glucose metabolism can be abrogated by metabolic stress [21,26]. Decreases in substrate supply, such as glucose or oxygen deprivation, or increase in energy demand will increase AMP/ATP ratios and subsequently activate adenosine monophosphate-activated protein kinase (AMPK) [27]. Changes in Ca^2+^ can also activate AMPK independently of AMP/ATP ratio [27,28]. Activated AMPK stimulates catabolic ATP-generating pathways and insulin-independent stimulation of glucose transport into the cardiomyocyte [29]. In hypoxic conditions, AMPK favors glycolysis by activating PFK-1 and PFK-2. Under normoxic conditions, AMPK activation favors FA oxidation by decreasing malonyl-CoA, an allosteric inhibitor of carnitine palmitoyl-transferase I (CPTI), and allowing transport of FA into the mitochondria [30]. Therefore, AMPK overrides the biochemical signals provided by the Randle cycle and glucose metabolism is not restricted by FA oxidation.

Hormonal or pharmacological stressors such as epinephrine can also induce a similar stress environment for the heart [31,32]. Epinephrine stimulates glycolysis in the heart by its second messenger, cyclic AMP (cAMP), and overrides metabolic control by FAs. cAMP increases glucose uptake, PFK flux, and PDH activity. Furthermore, epinephrine promotes FA utilization as a result of acetyl-CoA carboxylase (ACC) inactivation by cAMP protein kinase. With epinephrine, the heart utilizes both glucose and FA, but carbohydrates are largely metabolized to sustain the increase in heart function [21]. FA oxidation can also be inhibited by glucose. As glucose is metabolized through glycolysis and enters the TCA cycle as citrate, it can escape oxidation and enter the cytosol [33]. Some citrates regenerate into acetyl-CoA and are subsequently carboxylated to malonyl-CoA through ACC. Malonyl-CoA prevents the transport of long chain fatty acids (LCFA) into the mitochondria. The inhibition of LCFA oxidation also stimulates glucose uptake in cardiomyocytes.

FAs can also exert long-term effects through transcriptional regulation [34]. Certain FAs can bind to peroxisome proliferator-activated receptors (PPARs) and regulate lipid metabolism. PPARs act on multiple pathways to both store and oxidize lipids with a capacity to prevent oxidative stress [24,35]. The PPAR family has three members, α, -β, and -γ, and are all expressed in the heart [14]. Although PPARα is the dominant isoform in the heart, PPARβ and γ can also modulate FA metabolism [36]. PPARα stimulates the transcription of lipoprotein lipase, involved in FA uptake and β-oxidation in the mitochondria [37,38,39,40]. In addition, overexpression of PPARα in the heart favors FA metabolism and induces insulin resistance [41]. Although PPARβ is ubiquitously expressed, both PPARα and β increase PDK4 mRNA and decrease glucose oxidation [42]. The upregulation of PDK can be a response to high-fat diet or starvation and contributes to metabolic inflexibility associated with early stages of cardiac dysfunction [22,43]. Estrogen-related receptor (ERR) α shares common targets with PPARα and stimulates the transcription of genes involved in both glucose and oxidative phosphorylation [44]. The effects of PPARs and ERRs are potently activated by interacting with the PPARγ coactivator-1 (PGC-1) family. PGC-1 coactivates PPARα and ERRα to significantly increase FA uptake and oxidation and it is also a potent activator for mitochondrial biogenesis [45]. Interestingly, AMPK can also increase ATP production by activating PGC-1 to support the increase in FA and glucose oxidation [46].

### 2.3. The Mitochondrial Shuttles

The reduced pyridine nucleotides NADH and FADH_2_ are produced in the mitochondria through the TCA cycle and FA oxidation to deliver electrons to the respiratory chain. The exergonic flow of electrons is coupled to the transfer of protons across the inner mitochondrial membrane. This process converts the energy obtained from the oxidized substrates into both an electrical and chemical potential, which together is referred to as the proton motive force. Under physiological conditions, the proton motive force drives ADP phosphorylation by ATP synthase [47]. Similar to the function of AMP in the cytosol, ADP stimulates ATP synthase activity to accelerate electron flux and generate more ATP. Increases in mitochondrial Ca^2+^ can activate the TCA cycle to regenerate NAD^+^ and FAD and subsequently supply substrates to replenish NADPH, which activates antioxidants such as glutathione peroxidase and glutaredoxin. However, phosphorylation of NADH and FADH_2_ generates reactive oxygen species (ROS) mainly through complexes I and III [48] and can have a wide variety of cellular processes, ranging from protective mechanisms such as preconditioning to deleterious effects such as oxidative damage and ventricular remodeling [49].

FA and glucose do not freely cross the mitochondrial membrane but require shuttle systems. At the end of glycolysis in the cytosol, pyruvate is transported into the mitochondria through voltage-dependent ion channels for the outer mitochondrial membrane and the mitochondrial pyruvate carrier (MPC) for the inner mitochondrial membrane. Controlled by its bioavailability and post-translational modifications, MPC activity can determine whether pyruvate is oxidized in the mitochondria or in the cytosol [50]. FAs in the cytosol are first esterified to fatty acyl-CoA and then either esterified to triglyceride for storage or converted to long-chain acylcarnitine by CPTI for oxidation [51]. While the transport of long-chain fatty acids is dependent on the bioavailability of carnitine, short- and medium-chain fatty acids are carnitine-independent and can freely enter the mitochondria [52]. However, short- and medium-chain FAs mostly serve as a modulator of metabolism by inhibiting glycolysis, instead of being oxidized for energy. Long-chain acylcarnitines are shuttled into the mitochondria matrix and then converted back to long-chain acyl-CoA by carnitine palmitoyl-transferase II (CPTII). In the mitochondrial matrix, the long-chain acyl-CoA can undergo β-oxidation. ATP is transferred out of the mitochondria through the creatine kinase buffer system or directly through ADP/ATP carriers. Creatine kinase transfers the high energy phosphate bond in ATP to creatine and forms phosphocreatine. Phosphocreatine can rapidly diffuse to the cytosol and transfer the phosphoryl group to ADP and create ATP. The molecular mechanism of the ADP/ATP carrier in the mitochondria was recently elucidated where the carrier imports one ADP and exports one ATP from the mitochondria based on concentration gradients [53].

## 3. Metabolism in the Failing Heart

The fundamental stages of energy metabolism that are perturbed in HF include substrate uptake and selection, oxidative phosphorylation, and energy shuttling [9]. From human heart biopsies, ATP is 25–30% lower in the failing heart, resulting in ~3.5 μmol of ATP per gram of wet weight [15,54]. Consequences of sustained changes in the failing heart leads to a significant decrease in VO_2_ max to <14 mL/kg/min [55]. The heart also suffers a metabolic change from predominantly metabolizing FA to glucose [56]. This metabolic switch is initially beneficial and compensatory, but will ultimately result in insulin insensitivity and a loss of metabolic flexibility [57,58]. The decrease in FA oxidation can also be attributed to the suppression of PPARα, which impairs FA transport into the mitochondria [59,60,61,62]. Heart failure can be associated with increased circulating FAs due to high lipolysis rates [63,64,65]. While FAs are not oxidized in the mitochondria, cellular FA uptake is not decreased. The accumulation of FA in the cytosol, however, can lead to non-oxidative pathways producing lipotoxic species such as ceramide and diacylglycerol [66]. Subsequently, the lipotoxic species can lead to further mitochondrial dysfunction and apoptosis [67,68]. The accumulation of FA in the cytosol can also induce insulin resistance through post-translational modifications of the insulin signaling cascade [69]. Interestingly, the expression of proteins involved in FA omega-oxidation, a rescue pathway for FA oxidation in the endoplasmic reticulum, has not been observed in HF [70,71].

While FA oxidation is decreased, glucose uptake and glycolysis is significantly augmented in the failing heart [72,73,74]. This is likely due to FA oxidation being 11–12% less oxygen efficient than glucose oxidation [10,75]. Despite the switch towards glucose metabolism, the failing heart remains energy deficient due to the change in pyruvate oxidation. Instead of being transported into the mitochondria for oxidation, pyruvate remains in the cytosol and is reduced to lactate [76,77]. This phenomenon can be partially attributed to the downregulation of MPC [78]. Pyruvate oxidation also changes in the mitochondria of failing hearts. Pyruvate may be shuttled to anaplerotic pathways where pyruvate can directly enter the later stages of the TCA cycle through pyruvate carboxylation by malic enzymes [79]. This late entry in the TCA cycle is not sufficient to compensate for the impaired FA oxidation. In the late stages of HF, glucose uptake decreases due to insulin insensitivity. This induces PPARα activation for mitochondrial biogenesis, but excessive PPARα activation results in contractile dysfunction [45]. To make up for the depleted cellular energetics, ketone bodies and branched amino acids (BCAA) become more relevant in the failing heart [80,81]. Ketone bodies provide a more efficient ATP to oxygen ratio than palmitate [82,83]. However, it remains unclear whether relying on ketone bodies is beneficial. Current literature is inconclusive on whether metabolic dysfunction is the result of contractile dysfunction or vice versa.

## 4. Cardiovascular Metabolomics

Traditional research approaches have relied on a reductionist methodology where metabolites and pathways are studied under a narrow scope. The development of -omics platforms, including genomics, transcriptomics, proteomics, and metabolomics provide an unbiased and holistic view of potential metabolic changes that can occur in biological systems. Metabolomics has been gaining popularity with its extensive capabilities to measure chemical intermediates and metabolites in a variety of biological samples. This platform is rapidly expanding our understanding of metabolism in different disease states [84]. Furthermore, metabolomics integrates genomic, epigenetic, transcriptomic, and proteomic variation, while also incorporating environmental factors such as environmental exposures, physical activity, diet, and the microbiota. The metabolic characterization of the failing heart, a complex pathology with genetic and environmental contributions, has provided important insight in its pathogenesis and identifying new therapeutic targets and biomarkers.

There are several methods to investigate metabolomics, but their comprehensive review is beyond the scope of this article. For simplicity, this article will review nontargeted and targeted metabolomics. Nontargeted metabolomics offers a comprehensive approach to identify and relatively quantify the most abundant metabolites in a biospecimen. To maximize the number of metabolites captured, this method loses the granularity associated with genotype, pharmacological treatment, or other comparative groupings. Nontargeted metabolomics is limited by metabolite identification, which requires an exhaustive metabolite database, biochemical modifications of metabolites that result in spectral shifts, and the dearth of comprehensive fragmentation chemistries for all metabolites. Nonetheless, the nontargeted approach offers a global perspective of the tissue’s metabolism and has the potential to identify novel biomarkers and metabolic pathways. In comparison, targeted metabolomics offers the ability to obtain absolute quantification of a selective group of metabolites. The major benefits of this method are the increased selectivity and sensitivity due to metabolite extraction and compound separation. Instrument parameters can be tailored to specific metabolite panels and quantification is achieved by including stable isotope-labeled internal standards or calibration curves with an external standard.

### Heart Failure

Metabolic changes often precede cardiac functional changes in the failing heart suggesting that metabolic reprogramming is an early marker of HF pathogenesis [85]. Thus, investigations over the past several decades have been focused on changes in FA and glucose. This approach has generally neglected other metabolic pathways in the failing heart, whereas unbiased molecular profiling can capture the discovery of previously unappreciated pathways that contribute to HF etiology.

A study in mice assessing metabolic changes associated with infarct-related HF found changes in 40% of the 288 measured metabolites [86]. There were also time-dependent decreases in metabolites such as purines, acylcarnitines, and FAs suggesting global effects in energy metabolism. Furthermore, there was an observed accumulation of BCAA, which was associated with insulin resistance. Several additional investigations were conducted using metabolomics to elucidate the metabolism of BCAA in HF. Integrated transcriptomics and metabolomics were utilized in a HF mouse model and revealed a coordinated downregulation of the BCAA catabolic pathway in the cell and accumulation in the tissue and in circulation [87]. These observations were also found in human HF cardiac and plasma samples. Application of therapeutics to enhance BCAA catabolism in mice resulted in delayed HF progression, suggesting that downregulation of the cardiac BCAA pathway may contribute to HF. Another lipodomic and targeted metabolomics study of human HF myocardium samples showed evidence of impaired FA oxidation and increased downstream metabolites and enzymes of the ketone oxidation pathway [80].

Human myocardial tissue samples reflecting early disease are difficult to obtain and thus, most human HF metabolomic studies have relied on end-stage tissue or serum/plasma samples [88]. Results from peripheral blood reflect the integrated systemic metabolism during HF rather than myocardial metabolism. HF causes widespread systemic changes in metabolism, rather than direct metabolic changes in the heart [89]. Interestingly, targeted metabolomics in the HF-ACTION (Heart Failure: A Controlled Trial Investigating Outcomes of Exercise Training) trial of patients with HF with reduced ejection fraction (HFrEF) revealed that circulating long-chain acylcarnitines were significantly associated with worsening cardiovascular outcomes [90]. The importance of long-chain acylcarnitines was further elucidated when it was significantly increased in individuals with end-stage HFrEF and significantly decreased after ventricular unloading with mechanical support [90]. Circulating long-chain acylcarnitines have also been shown to increase as the severity of HFrEF progresses [91,92]. While the mechanistic cause of circulating long-chain acylcarnitines remain unclear, these findings reflect systemic metabolic perturbations in FA oxidation and mitochondrial dysfunction [93].

## 5. Metabolic Targets for Pharmacological Treatment of Heart Failure

The significance of metabolic alterations in cardiac health has drawn interest in the pharmaceutical industry. Currently, most HF treatments target neuroendocrine pathways. These pathways may overlap with metabolic pathways (Figure 2), but often the mechanism is not well understood.

### 5.1. β-Blocker

*β*-blockers are one of the oldest therapies for HF. Currently randomized controlled trials have found that carvedilol, bisoprolol and metoprolol CR/XL reduce morbidity and mortality in patients with HFrEF [94]. While *β*-blockers’ beneficial effect in HF is often thought to be secondary to its sympatholytic activity resulting in decreased heart rate, decreased RAS activation, and reduced cardiac remodeling, *β*-blockers may also aid the failing heart through metabolic mechanisms. One of the likely mechanisms in which *β*-blockers target cardiac metabolism is by inhibition of CPTI activity (Figure 2A) [95]. This results in increased glucose oxidation and more efficient oxygen use for ATP production [96]. Furthermore, *β*-blockers have been found to inhibit FA oxidation [97], which may be beneficial in the failing heart as shifting from FA oxidation to glucose oxidation is more oxygen efficient [7]. Studies in diabetic rats have shown that both acute and chronic *β*-blocker therapy results in decreased FA oxidation in failing cardiac tissue through reduction in total CPTI activity [98]. This led to stimulation of glucose oxidation and increased tissue ATP levels in cardiac tissue [98].

Lastly, several studies have suggested that *β*-blockers may benefit the failing heart by reducing oxidative stress [99]. A potential mechanism in which *β* -blockers decrease oxidative stress is through preventing damage to cell membranes through lipid peroxidation [100,101,102]. Lipid peroxidation results in the formation of toxic aldehydes such as 4-hydroxy-2-nonenal (HNE) [101,103]. HNE inactivates proteins and DNA in cardiomyocytes by forming chemical adducts and as a result of this directly depresses contractile function [104]. Endomyocardial biopsies from patients with dilated cardiomyopathy revealed significantly increased expression of HNE modified proteins compared to healthy controls [105]. Studies have also shown that HF patients have elevated serum HNE compared to healthy controls and that HNE levels are inversely correlated with left ventricle contractile function [106].

While there are several mechanisms in which *β*-blockers as a class reduce ROS, there is substantial evidence that carvedilol specifically has direct antioxidant properties [107,108,109]. Carvedilol is a β_1_, β_2_ and α_1_ adrenergic blocking agent with evidence of effect in HF. Carvedilol is a chemical antioxidant and able to bind and scavenge potent ROS including the O**_2_^•^** radical as well as biologically prevent the production of ROS [110]. Administration of carvedilol was shown to significantly decrease the amount of HNE modified proteins in failing hearts [105]. Furthermore, while the true physiological cause of the different outcomes can never be fully elucidated, the COMET trial found that treating HF patients with carvedilol compared to metoprolol significantly decreased mortality [111].

*β*-blockers may further benefit the failing heart by reducing the amount of FA available to cardiomyocytes. This is beneficial as FA can be metabolized into intermediates such as diacylglycerol, which in mice have been shown to potentially contribute to cardiac insulin resistance and reduce cardiac function [112]. While there are several metabolic pathways that may be targeted by *β*-blockers, most of these pathways have been studied in rats and the translatability to humans is unknown.

### 5.2. ACE Inhibitor and ARB

While our understanding of HF has evolved significantly over time, the 1980s was a time of breakthrough in HF treatment when scientists and physicians no longer viewed HF as simply a failing pump, but a complex neuroendocrine condition [113]. The renin-angiotensin system (RAS) is physiologically important for homeostasis through the modulation of vascular tone, volume status and electrolyte balance. However, during HF, inappropriate activation of RAS can lead to cardiac remodeling and end organ damage [114]. Angiotensinogen, the initial component of RAS, is converted into angiotensin I by renin as the rate limiting step of RAS [115]. Renin is released by several stimuli including renal baroreceptor stimulation of low perfusion, changes in sodium delivery to the macula densa, sympathetic stimulation and inhibitory feedback by downstream molecules [116]. Angiotensin I is then converted to angiotensin II (ang II) by membrane bound angiotensin converting enzyme (ACE). Ang II binds to at least four angiotensin receptors; the most clinically relevant is the type 1 (AT1) receptor. Ang II acts on the AT1 receptor, which is responsible for vasoconstriction, cardiac fibrosis, hypertrophic remodeling, renal sodium retention and aldosterone synthesis in the adrenal glands [116,117]. Furthermore, ang II promotes endothelial cell dysfunction through dysregulation of nitric oxide signaling [114,117,118,119].

With this evolution in the understanding of HF, ACE inhibitors became a mainstay treatment. The landmark CONSENSUS trial (Effects of Enalapril on Mortality in Severe Congestive Heart Failure) demonstrated survival and symptom benefits of ACE inhibitors in patients with New York Heart Association (NYHA) class IV HF [120]. In the same vein, angiotensin receptor blockers (ARBs) block AT1 without some of the limiting side effects seen in ACE inhibitors [116,121,122]. Much of the benefit from ACE inhibitors and ARBs is thought to be their action in the RAS. However, ACE inhibitors and ARBs can also act on metabolic pathways in HF (Figure 2B).

Ang II damages mitochondria in the failing heart through generation of reactive oxygen species [123]. A study found that mice with HF treated with ACE inhibitors exhibited downregulation of metabolic pathways involved in generating reactive oxygen species and low-density lipoprotein pathways [124]. Furthermore, ang II likely reduces glucose oxidation, acetyl-CoA, and ATP by upregulating PDK4 expression and this was prevented by an ARB [125]. Further studies revealed ang II-induced increase in PDK4 expression may further exacerbate ATP depletion in the failing heart by promoting insulin resistance and causing cardiomyocytes to utilize FAs instead of glucose for energy [126]. While it is unclear to what degree the changes in metabolism is attributable to the utilization of ACE inhibitor or ARB, there are several clear metabolic targets for these treatments.

### 5.3. Angiotensin Receptor-Neprilysin Inhibitor (ARNI)

The demonstration of the combination of an ARB (Valsartan) with sacubitril, a neprilysin inhibitor, to significantly lower mortality in the PARADIGM-HF trial (Angiotensin–Neprilysin Inhibition versus Enalapril in Heart Failure) has recently changed the dogma in the treatment of patients with heart failure [127]. Neprilysin is a peptidase that catalyzes the degradation of a number of vasodilator peptides, most importantly natriuretic peptides [128]. Neprilysin also catalyzes the degradation of ang II [129]. Natriuretic peptides, including atrial natriuretic peptide (ANP) found in the atria and brain natriuretic peptide (BNP) found in the ventricles, are cardioprotective as they promote diuresis, vasodilation, and inhibit renin secretion [130].

Current American Heart Association (AHA) guidelines recommend initiation of ARNIs for patients with HFrEF with AHA class C [131]. The PARAGON-HF trial did not show reduced hospitalization or death in patients with heart failure with preserved ejection fraction (HFpEF) [132]. Beyond natriuretic peptide’s well-known effects on the neuroendocrine system, several studies have suggested that natriuretic peptides are also metabolically active. ANP is known to stimulate lipolysis [133] and multiple studies have shown neprilysin inhibitor benefit in glycemic control in type 2 diabetes [134,135]. A potential mechanism for which neprilysin inhibitors improve insulin sensitivity is by preventing proteolysis of glucagon-like peptide-1 (GLP-1) [136]. Neprilysin inhibitors’ action on GLP-1 may also contribute to their beneficial effect in HF as GLP-1 receptors are expressed on the heart. Several animal studies have shown that increasing GLP-1 improves survival and cardiac function through increased cardiac glucose uptake [137,138]. The extent of the protective effect of GLP-1 in humans is still inconclusive. While randomized controlled trials have shown GLP-1 agonists to be neutral for HF, a recent meta analysis of GLP-1 agonists in patients with type 2 diabetes mellitus found a small reduction in hospitalization secondary to HF [139].

### 5.4. Sodium-Glucose Transport Protein 2 (SGLT2) Inhibitors

SGLT2 inhibitors are one of the newest classes of treatments for HF with one of the strongest degrees of evidence for a potential metabolic mechanism of action. Multiple studies have shown benefit of SGLT2 inhibitors in patients with type 2 diabetes mellitus and HF [140,141,142]. In fact, the DAPA-HF (Dapagliflozin in Patients with Heart Failure and Reduced Ejection Fraction) demonstrated decreased HF morbidity and mortality in patients regardless of diabetes status on an SGLT2 inhibitor [143]. Currently the AHA recommends initiating a SGLT2 inhibitor in all patients with HFrEF with NYHA Class II- IV disease with adequate kidney function [131]. The groundbreaking EMPEROR-Preserved trial found that SGLT2 inhibitors reduce the combined risk of hospitalization and cardiovascular death in patients with HFpEF regardless of diabetes status, introducing a new treatment to an illness with previously few treatment options [144].

There are a number of mechanisms in which SGLT2 inhibitors are thought to improve cardiac function in HF, including promoting diuresis, preventing cardiac remodeling, improving glycemic control, and preventing sympathetic nervous system activation [145]. In addition, SGLT2 inhibitors may benefit a failing heart by increasing ketones and free FAs as an energy source and decreasing mitochondrial ROS (Figure 2B).

As described previously, a failing heart is reliant on glycolysis as mitochondrial oxidative processes decline. As the mitochondria fail, glucose oxidation decreases, resulting in an energy depleted heart [145]. SGLT2 inhibitors mobilize adipose tissue FAs, which are converted to ketones in the liver [146]. Studies suggest that ketones can be useful as a fuel source for failing hearts [80,147], but this is controversial. Animal HF models have shown that SGLT2 inhibitors improve cardiac function by increasing ketones, free FAs, and BCAA [148,149]. Another mechanism in which SGLT2 inhibitors act on metabolic pathways in HF is their effect on mitochondrial ROS [7]. During HF, increased oxidative stress can result in mitochondrial dysfunction and cardiac remodeling [150]. Studies in diabetic mice have shown that administration of SGLT2 inhibitors reduces oxidative stress to the heart and cardiac fibrosis by promoting nuclear translocation of Nrf2, an essential signaling component in oxidative pathways [151]. While SGLT2 inhibitors improve cardiac function in HF by many mechanisms, there is increasing evidence that regulation of cardiac metabolism is an important component.

### 5.5. Therapies Currently under Investigation

While many of the standard therapies in HF have been found to act on the metabolic pathways of cardiomyocytes, there are several therapies that primarily target cardiac metabolism that are under investigation.

Trimetazidine is a partial inhibitor of long-chain 3-ketoacyl CoA thiolase, the key enzyme in the β-oxidation pathway. It is the only therapy that directly targets this metabolic pathway. Inhibition of long-chain 3-ketoacyl CoA shifts myocardial metabolism from FA oxidation to glucose oxidation, which as previously discussed may be beneficial in increasing energy substrate to the failing heart [96]. Currently, the 2019 European Society of Cardiology has a level IIa recommendation for the use of trimetazidine in stable coronary syndrome [152]. Several randomized controlled trials and a meta-analysis of these trials have shown benefit of trimetazidine in HF [153,154]. Randomized controlled trials have also shown benefit of Trimetazidine in decreasing insulin resistance in patients with heart disease, which may be another mechanism in which they promote myocardial health [155,156].

Another potential treatment for HF under investigation that targets the metabolic pathways is malonyl coenzyme A decarboxylase (MCD) inhibitors. MCD inhibitors work by increasing cardiac malonyl coenzyme A levels. This causes inhibition of CPTI, which results in reduced mitochondrial FA uptake and favoring of glucose oxidation pathways [96]. There is also evidence that MCD inhibitors act on pathways to minimize ROS. Animal studies have shown promise as MCD inhibitors are able to reverse HF [157,158]. However, there are concerns that MCD inhibitors could cause liver steatosis and may not be suitable for this reason [96]. However, in rodents, MCD inhibitors did not have this problem possibly because malonyl CoA is more sensitive in inhibiting FA oxidation in heart versus liver mitochondria [157]. While this is an active area of investigation, there are no MCD inhibitors available on the market.

Fibrates are another medication that targets heart metabolism that is currently under investigation for HF. Fibrates are PPARα and PPARγ agonists, which act by decreasing the circulating free FA supply to the heart [96]. Decreased FA supply to the heart reduces cardiac FA oxidation rates and increases glucose oxidation rates, which is initially beneficial in the ischemic heart [123]. The clinical translation of this metabolic theory is still up for debate as clinical trials have mostly shown no benefits of fibrates in cardiovascular disease [159,160], but there is growing evidence that fibrates can be useful for patients with atherogenic dyslipidemia [161].

## 6. Conclusions

During HF, the metabolic pathways of the heart become perturbed. A metabolic switch occurs, from glucose oxidation to FA oxidation, which results in damaging oxidative distress and a paucity of energy for a heart with further energetic demands. Many of the standard HF treatments, developed for their action on RAS, also target the metabolic pathways that become deranged in HF. While HF is a complex physiologic process, understanding of the pathophysiology is incomplete without a discussion of metabolic derangement. It is important to consider the development of other promising treatments that target metabolic pathways.

## Figures and Tables

**Figure 1 ijms-23-02960-f001:**
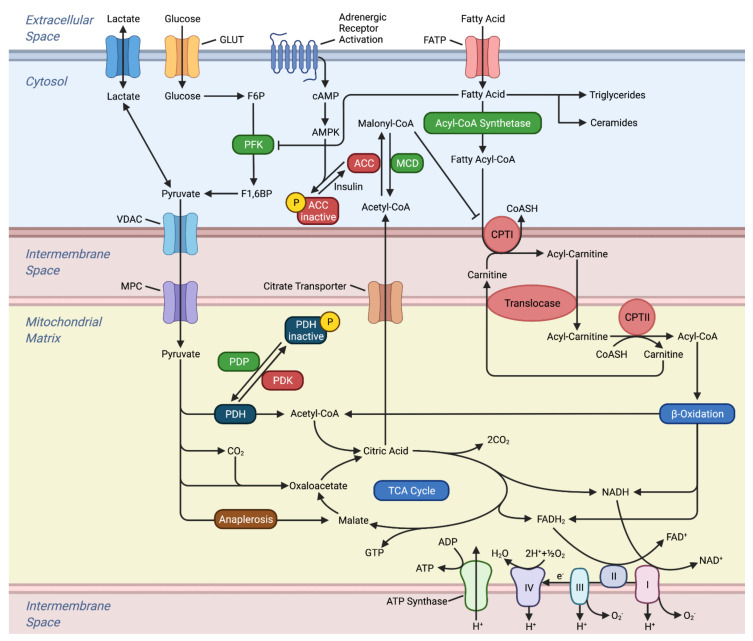
Metabolism in the Healthy Heart. This figure illustrates the oxidation and transport of glucose and fatty acids in the heart. The electron carriers generated from β-oxidation and the TCA cycle, which subsequently enters the electron transport chain. Abbreviations, ACC: acetyl-CoA carboxylase, ADP: adenosine diphosphate, AMPK: adenosine monophosphate activated protein kinase, ATP: adenosine triphosphate, cAMP: cyclic AMP, CPTI: carnitine palmitoyl-transferase I, CPTI: carnitine palmitoyl-transferase II, CoASH: coenzyme A, FATP: fatty acid transport protein, F1,6BP: fructose 1,6-bisphosphate, F6P: fructose 6-phosphate, GLUT: glucose transporter, MCD: malonyl coenzyme A decarboxylase, MPC: mitochondrial pyruvate carrier, PDH: pyruvate dehydrogenase, PDK: pyruvate dehydrogenase kinase, PDP: pyruvate dehydrogenase phosphatase, TCA: tricarboxylic acid, VDAC: voltage-dependent anion channel.

**Figure 2 ijms-23-02960-f002:**
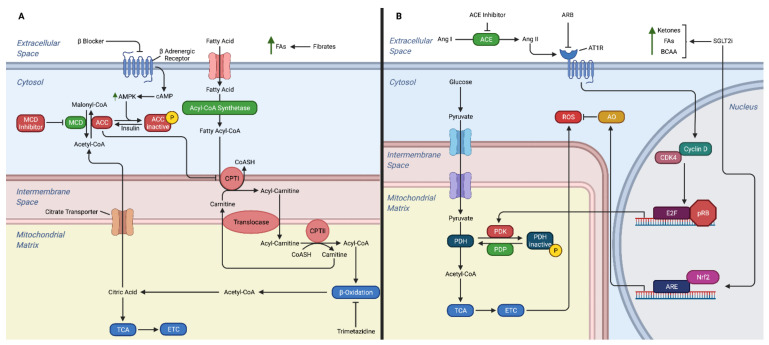
Mechanisms of action for heart failure medications on cardiac metabolism. (**A**) Fatty acid metabolism. (**B**) Glucose metabolism. The drugs shown in part A include β-blockers, fibrates, MCD inhibitor, and trimetazidine. Section B demonstrates the mechanisms of heart failure drugs on glucose metabolism and response to the production of ROS. The drugs shown in section B include ACE inhibitor, ARB, and SGLT2i. Abbreviations, ACE: angiotensin-converting enzyme, Ang: angiotensin AMPK: adenosine monophosphate activated protein kinase, AO: antioxidants, ARB: angiotensin receptor blockers, ARE: antioxidant response element, AT1R: angiotensin II type 1 receptor, BCAA: branched chain amino acids, cAMP: cyclic AMP, CDK: cyclin dependent kinase, CPTI: carnitine palmitoyl-transferase I, CPTI: carnitine palmitoyl-transferase II, CoASH: coenzyme A, ETC: electron transport chain, FAs: fatty acids, MCD: malonyl coenzyme A decarboxylase, Nrf2: nuclear erythroid 2-related factor 2, PDH: pyruvate dehydrogenase, PDK: pyruvate dehydrogenase kinase, PDP: pyruvate dehydrogenase phosphatase, pRB: product of the retinoblastoma tumor suppressor gene, ROS: reactive oxygen species, SGLT2i: sodium-glucose transport protein 2 inhibitor, TCA: tricarboxylic acid.

**Table 1 ijms-23-02960-t001:** Heart failure therapies and mechanism of action.

Therapy	Action in RAS	Proposed Metabolic Mechanism
ACE inhibitor	Inhibit conversion of Ang I to Ang II preventing vasoconstriction and aldosterone release	Prevent Ang II mediated increased PDK4 expression resulting in increased glucose oxidation.
ARB	Competitive antagonist of AT1 receptor, preventing vasoconstriction and aldosterone release	Prevent Ang II mediated increased PDK4 expression resulting in increased glucose oxidation.
ARNI	Mechanism of ARB with combined neprilysin inhibition preventing break down of natriuretic peptides	Increased ANP stimulating lipolysis and preventing GLP-1 proteolysis, although mechanism and benefit not fully elucidated. Further addition of ARB mechanism of action.
*β*-Blocker	Protective from *β*-1 adrenoreceptor overstimulation causing tachycardia, ventricular remodeling and vasoconstriction	Inhibition of CPTI activity, resulting in inhibition of FA oxidation, causing cardiac metabolism to shift to more oxygen efficient forms of metabolism.
SGLT2 inhibitor	Diuresis effect on the kidney. Main mechanism of action not on RAS.	Increases ketone genesis in the liver for the failing heart and reduces oxidative stress by promoting nuclear translocation of Nrf2.
Trimetazidine	No major mechanism in RAS	Partial inhibition of long-chain 3-ketoacyl CoA thiolase shifts myocardial metabolism from FA oxidation to glucose oxidation.
MCD inhibitors	No major mechanism in RAS	Increase cardiac malonyl coenzyme A levels, causing inhibition of CPTI, which results in reduced mitochondrial FA uptake and increased glucose oxidation pathways. May also minimize ROS. Role in HF therapy uncertain.
Fibrates	No major mechanism in RAS	PPARα and PPARγ agonists, which act by decreasing the circulating free FA supply to the heart. Role in HF therapy uncertain.

## Data Availability

Not applicable.

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
