# Peer review of "Heart Failure and Drug Therapies: A Metabolic Review"

_ijms, 2022, doi:10.3390/ijms23062960_

Round 1

Reviewer 1 Report

This article provides a broad overview of cardiac metabolism and its alterations in 'heart failure' along with a summary of current therapies, their benefits, and evidence of the same mediated by modulation of cardiac metabolism. The strength of this review is largely in Section 5 where heart failure therapeutics are discussed. A major weakness is the lack of quantitative physiological context of cardiac function and dysfunction - 1) absence of ranges of fluxes of respiration and ATP consumption in normal and failing hearts, 2) lack of references to the physiological operating point whenever changes in substrate utilization or other factors are discussed, 3) lack of linkage to clinical staging of cardiac dysfunction (AHA/ACC) when feasible although references to end stage HF are provided. In view of this, my suggestion to the authors is to incorporate more specific aspects of cardiac metabolic function beyond the generic descriptions, which are present in innumerable prior review articles on this topic. Then, elaborate on the strongest aspect of this article - Section 5 on pharmacological targets, which is a unique contribution to the reviews in this field. 

Minor point - Section 5.3, line 323 "Neprilysin catalyzes ang II" should be corrected to "Neprilysin catalyzes degradation of ang II" (?)  

Author Response

Reviewer 1 Comments:

This article provides a broad overview of cardiac metabolism and its alterations in 'heart failure' along with a summary of current therapies, their benefits, and evidence of the same mediated by modulation of cardiac metabolism. The strength of this review is largely in Section 5 where heart failure therapeutics are discussed.

A major weakness is the lack of quantitative physiological context of cardiac function and dysfunction -

1) absence of ranges of fluxes of respiration and ATP consumption in normal and failing hearts,

2) lack of references to the physiological operating point whenever changes in substrate utilization or other factors are discussed,

3) lack of linkage to clinical staging of cardiac dysfunction (AHA/ACC) when feasible although references to end stage HF are provided.

Response: Thank you for suggesting that our paper would be strengthened with additional quantitative data including references to physiological and deranged metabolic levels. While we do agree with this comment, we also believe that the quantification of ATP in failing hearts would strongly depend on the underlying etiology of the heart failure and existing comorbidities of the disease, which is beyond the scope of this paper. We focused this paper on the foundations of metabolism in heart failure to support the pharmacological targets section. We do believe that more quantitative information would be helpful for the readers and have made the changes listed below.

Page 6 Section 2.1; lines 73-74: “The energy pool of the heart consists of ~5 μmol of ATP per gram of wet weight [15].”

Page 9 Section 3; lines 204-205: “From human heart biopsies, ATP is 25-30% lower in the failing heart, resulting in ~3.5 μmol of ATP per gram of wet weight  [15, 53].”

Page 10 Section 3; lines 222-223: “This is likely due to FA oxidation being 11-12% less oxygen efficient than glucose oxidation [10, 73].“

Thank you for suggesting that addressing clinical staging of heart failure would benefit our review. We have addressed this by including indications for heart failure therapies based on AHA recommendations and heart failure classification (reduced versus preserved). We have also included more information about landmark trials that inform these guidelines.

In view of this, my suggestion to the authors is to incorporate more specific aspects of cardiac metabolic function beyond the generic descriptions, which are present in innumerable prior review articles on this topic. Then, elaborate on the strongest aspect of this article - Section 5 on pharmacological targets, which is a unique contribution to the reviews in this field.

Response: Thank you for suggesting that our review would benefit from expansion on section 5. To address this comment, we elaborated the role of beta blockers in decreasing oxidative stress in heart failure in section 5.1. We also expanded the section with regard to the guidelines for HF treatment as described above.

We added the following additions to the revised manuscript:

Page 13 Section 5.1; lines 327-329: “Currently randomized controlled trials have found that carvedilol, bisoprolol and metoprolol CR/XL reduce morbidity and mortality in patients with HFrEF [92].”

Page 13 Section 5.1; lines 341-361: “Lastly, several studies have suggested that β -blockers may benefit the failing heart by reducing oxidative stress [142]. A potential mechanism in which β -blockers decrease oxidative stress is through preventing damage to cell membranes through lipid peroxidation [97–99]. Lipid peroxidation results in the formation of toxic aldehydes such as 4-hydroxy-2-nonenal (HNE) [98, 100]. HNE inactivates proteins and DNA in cardiomyocytes by forming chemical adducts and as a result of this directly depresses contractile function [101]. Endomyocardial biopsies from patients with dilated cardiomyopathy revealed significantly increased expression of HNE modified proteins compared to healthy controls [102]. Studies have also shown that HF patients have elevated serum HNE compared to healthy controls and that HNE levels are inversely correlated with left ventricle contractile function [103].

While there are several mechanisms in which β-blockers as a class reduce ROS, there is substantial evidence that carvedilol specifically has direct antioxidant properties [104–106]. Carvedilol is a β1, β2 and α1 adrenergic blocking agent with evidence of effect in HF. Carvedilol is a chemical antioxidant and able to bind and scavenge potent ROS including the O2• radical as well as biologically prevent the production of ROS [107]. Administration of carvedilol was shown to significantly decrease the amount of HNE modified proteins in failing hearts [102]. Furthermore, while the true physiological cause of the different outcomes can never be fully elucidated, the COMET trial found that treating HF patients with carvedilol compared to metoprolol significantly decreased mortality [108].”

Page 14 Section 5.2; line 391: “The landmark CONSENSUS trial (Effects of Enalapril on Mortality in Severe Congestive Heart Failure) demonstrated survival and symptom benefits of ACE inhibitors in patients with New York Heart Association (NYHA) class IV HF [117].”

Page 15 Section 5.3; line 415: “Neprilysin is a peptidase that catalyses the degradation of a number of vasodilator peptides, most importantly natriuretic peptides [125]. Neprilysin also catalyses the degradation of ang II [126].”

Page 15 Section 5.3; lines 419-422: “Current American Heart Association (AHA) guidelines recommend initiation of ARNIs for patients with HFrEF with AHA class C [128]. The PARAGON-HF trial did not show reduced hospitalization or death in patients with heart failure with preserved ejection fraction (HFpEF) [129].”

Page 15-16 Section 5.4; lines 443-448: “Currently the AHA recommends initiating a SGLT2 inhibitor in all patients with HFrEF with NYHA Class II- IV disease with adequate kidney function [128]. The groundbreaking EMPEROR-Preserved trial found that SGLT2 inhibitors reduce the combined risk of hospitalization and cardiovascular death in patients with HFpEF regardless of diabetes status, introducing a new treatment to an illness with previously few treatment options [142].”

Minor point - Section 5.3, line 323 "Neprilysin catalyzes ang II" should be corrected to "Neprilysin catalyzes degradation of ang II" (?) 

Response: Thank you for the minor point and we have made the appropriate in line changes.

Reviewer 2 Report

Frank Yu et al., highlighted cardiac metabolism importance during healthy and failing heart and current therapeutic drugs. Review article written well, however my concerns are... authors must have additional figure showing heart failure therapeutic targets including mechanism of action.

Reference styles need be corrected (refe no: 21, 26, 40, 47, 68, 71, 93, 89, 116, 117, 122)

Author Response

Reviewer 2 Comments:

Frank Yu et al., highlighted cardiac metabolism importance during healthy and failing heart and current therapeutic drugs.  Review article written well, however my concerns are...

Authors must have additional figure showing heart failure therapeutic targets including mechanism of action.

Response: Thank you for your suggestion to add the pharmaceutical therapies in a metabolic diagram. We have included a new diagram in our revision and have addressed the references provided.

We have added the following figure and figure legend:

Figure 2. Mechanisms of action for heart failure medications on cardiac metabolism

  1. Fatty acid metabolism. B: Glucose metabolism

The drugs shown in part A include β-blockers, fibrates, MCD inhibitor, and trimetazidine. Section B demonstrates the mechanisms of heart failure drugs on glucose metabolism and response to the production of ROS. The drugs shown in section B include ACE inhibitor, ARB, and SGLT2i.

Abbreviations, ACE: angiotensin-converting enzyme, Ang: angiotensin AMPK: adenosine monophosphate activated protein kinase, AO: antioxidants, ARB: angiotensin receptor blockers, ARE: antioxidant response element, AT1R: angiotensin II type 1 receptor, BCAA: branched chain amino acids, cAMP: cyclic AMP, CDK: cyclin dependent kinase, CPTI: carnitine palmitoyl-transferase I, CPTI: carnitine palmitoyl-transferase II, CoASH: coenzyme A, ETC: electron transport chain, FAs: fatty acids, MCD: malonyl coenzyme A decarboxylase, Nrf2: nuclear erythroid 2-related factor 2, PDH: pyruvate dehydrogenase, PDK: pyruvate dehydrogenase kinase, PDP: pyruvate dehydrogenase phosphatase, pRB: product of the retinoblastoma tumor suppressor gene, ROS: reactive oxygen species, SGLT2i: sodium-glucose transport protein 2 inhibitor, TCA: tricarboxylic acid.

Reference styles need be corrected (refe no: 21, 26, 40, 47, 68, 71, 93, 89, 116, 117, 122)

Response: These references have been corrected.

Round 2

Reviewer 1 Report

The authors have largely addressed my concerns. One point that remains to be addressed is to contextualize the metabolic descriptions around the physiological operating point, which means rest or a level of effort beyond rest. This would mean exercise at various effort levels that could be expressed in mets for example.

Author Response

Thank you for thoughtful comments.  We have addressed your comments in the following ways:

Reviewer 1 Comments:

The authors have largely addressed my concerns. One point that remains to be addressed is to contextualize the metabolic descriptions around the physiological operating point, which means rest or a level of effort beyond rest. This would mean exercise at various effort levels that could be expressed in Mets for example.

Response to reviewer 1:

Thank you for providing us feedback on the manuscript. We appreciate and recognize the utility of providing clinical descriptions of the healthy and failing heart to better contextualize the metabolic derangements. VO2 max has been shown to be a clinical relevant and prognostic measurement of the physiological operating point. As such, we have made the following changes to the revised manuscript:

Page 6 Section 2.1; lines 75-80: “The energy pool of the healthy heart consists of ~5 μmol of ATP per gram of wet weight and allows the heart to meet the cardiovascular demands of 50 mL/kg/min VO2 delivery during maximal exercise (VO2 max) [15, 16]. >95% of the heart’s ATP is generated from oxidative phosphorylation in the mitochondria, while the remaining 5% is from glycolysis and to a lesser extent, the tricarboxylic acid (TCA) cycle [17].”

Page 10 Section 3; lines 209-210: “Consequences of sustained changes in the failing heart leads to a significant decrease in VO2 max to <14 mL/kg/min [55].”

Additional Changes

We felt that updating the title of the manuscript may better reflect the novel contributions of review paper. With the permission of the editor, we would like to propose a change to the title:

From “The metabolic changes in the failing heart: A review”

to

“Heart Failure and Drug Therapies: A Metabolic Review”.
